# Quinoa Ameliorates Hepatic Steatosis, Oxidative Stress, Inflammation and Regulates the Gut Microbiota in Nonalcoholic Fatty Liver Disease Rats

**DOI:** 10.3390/foods12091780

**Published:** 2023-04-25

**Authors:** Lingyue Zhong, Wei Lyu, Zihan Lin, Jun Lu, Yanlou Geng, Lihua Song, Heng Zhang

**Affiliations:** 1School of Agriculture and Biology, Shanghai Jiao Tong University, Shanghai 200240, China; 2National Semi-Arid Agriculture Engineering Technology Research Center, Shijiazhuang 050051, China; 3CAS Center for Excellence in Molecular Plant Sciences, Chinese Academy of Sciences, Shanghai 201602, China; 4School of Life Sciences and Biotechnology, Shanghai Jiao Tong University, Shanghai 200240, China

**Keywords:** gut microbiota, high fat diet, NAFLD, quinoa intake

## Abstract

A long-term high-fat diet causes hepatic steatosis, which further leads to oxidative stress and inflammation. In this study, we firstly investigated the regulation effects of different amounts of quinoa on hepatic steatosis, oxidative stress, and inflammation of rats fed a high-fat diet, then the gut microbiota was dynamically determined. Sprague–Dawley (SD, male) rats were randomized into four groups: normal controls (NC, fed standard chow), model groups (HF, fed a high-fat diet), low quinoa intake (HF + LQ), and high quinoa intake (HF + HQ) groups, which were supplemented with 9% and 27% quinoa in the high-fat feed (equivalent to 100 g/day and 300 g/day human intake, respectively). The results showed that quinoa intake significantly inhibited the hepatomegaly and splenomegaly, ameliorated hepatic steatosis pathologically; effectively rescued the decrease in the activities of superoxide dismutase (SOD) and glutathione peroxidase (GSH-PX) and the increase in malondialdehyde (MDA). The levels of tumor necrosis factor-α (TNF-α), interleukin-10 (IL-10), transforming growth factor-β (TGF-β), and leptin in rats of two quinoa groups were close to those of the NC group. Besides, high quinoa intake significantly increased the relative abundance of *Akkermansia*, and low quinoa intake significantly increased the relative abundance of *Blautia* at the genus level. The relative abundances of *Blautia* and *Dorea* in rats in the HF + HQ group were lower than those in rats in the HF + LQ group. In addition, the relative abundances of *Clostridium* and *Turicibacter* of rats in the two quinoa intervention groups were lower than those of rats in the HF group after 12 weeks of intervention. In summary, quinoa exhibits a series of beneficial effects in the prevention of nonalcoholic fatty liver disease (NAFLD) and is suggested to be a component of a daily diet for the prevention of NAFLD.

## 1. Introduction

Nonalcoholic fatty liver disease (NAFLD) refers to liver fat accumulation—so-called hepatic steatosis at a level ≥ 5% of the liver area without the presence of other diseases. It is estimated that 25% of the global population has NAFLD [1]. The prevalence of the disease may continue to increase in the near future since the presence of the disease is correlated with obesity or overweight, which has been rising in recent years [2]. The spectrum of the disease may range from hepatic steatosis to nonalcoholic steatohepatitis (NASH), cirrhosis, or even hepatocellular carcinoma, without intervention. Therefore, NAFLD has become a global health problem.

The etiology of NAFLD disease is multifactorial (“multiple hit” hypothesis), which involves insulin resistance (IR), lipotoxicity, oxidative stress, inflammatory response, genetic and epigenetic factors, and gut microbiota dysbiosis, etc. [3,4]. To date, there are no approved drugs for the treatment of NAFLD, and strategies mainly depend on lifestyle modifications, physical exercise, and diet management. A series of studies have been exploring functional components in food that can alleviate the disease and obtained some positive effects of several classes of antioxidants, such as polyphenols, on the reversion of fatty liver [5].

Quinoa (*Chenopodium quinoa* Willd.), as a pseudocereal grain, has attracted considerable attention in recent years because of its high nutritional value. It is rich in micronutrients, including vitamins (such as B, C, and E, etc.) and minerals (such as K, Ca, and Zn, etc.). The proteins efficiency ratio (PER) of quinoa is particularly high due to the excellent balance of essential amino acids [6]. Moreover, quinoa is also abundant in phytochemicals such as saponins, phenolic compounds, terpenoids, betanins, and carotenoids, which strongly suggests that quinoa has health benefits, including reducing body mass index (BMI) and fasting triglyceride in post-menopausal women, and lowering of postprandial glucose responses [7,8], antioxidant activities in high fructose-fed rats [9], and alleviating the inflammatory cytokines level [10]. Regarding the effect of quinoa on hepatic steatosis, only a recent study investigated quinoa intake on reducing cholesterol in plasma and liver, lessening obesity-associated chronic inflammation, and preventing hepatic steatosis in obese db/db mice [11]. Therefore, more exploration about the benefits of quinoa, as well as the influence of its intake that can be used in the prevention of NAFLD, is still needed.

It is well known that gut microbiota dysbiosis contributes to the pathophysiology of metabolic syndromes, which include NAFLD, through the gut–liver axis [12]. Some studies have explored the influence of polysaccharides, proteins, and saponins from quinoa on the gut microbiota using in vitro systems or other animal models [13,14,15,16,17]. To introduce a more effective diet strategy for the prevention of NAFLD, we investigated the alleviating effect of different amounts of whole grain quinoa on hepatic steatosis and the dynamic influence of quinoa intake on the gut microbiota of rats fed a high-fat diet in the present study.

## 2. Materials and Methods

### 2.1. Materials

Quinoa was grown in Wulan county in the Qinghai-Tibet Plateau (the mountain dryland areas which has an altitude of 4000 m) and was provided by Qinghai Sanjiang Wotu Ecology Agriculture Technology Co., Ltd. (Qinghai, China).

The commercial assay kits for superoxide dismutase (SOD) (A001-3-2) and glutathione peroxidase (GSH-PX) (A005-1-2) activities and the levels of malondialdehyde (MDA) (A003-1-2), glutathione (GSH) (A006-2-1), triglyceride (TG) (A110-1-1), total cholesterol (TC) (A111-1-1), and nonesterified free fatty acids (NEFAs) (A042-2-1) were purchased from Nanjing Jiancheng Biological Co., Ltd. (Nanjing, China). Rat enzyme-linked immunosorbent assay (ELISA) kits for detections of serum tumor necrosis factor alpha (TNF-α, ELR-TNFα-1), leptin (ELR-Leptin-1), and interleukin-10 (IL-10, ELR-IL10-1) were purchased from Raybiotech (Atlanta, GA, USA), and a transforming growth factor-β1 (TGF β1) Rat ELISA Kit (BMS623-3) was purchased from eBioscience (San Diego, CA, USA). The standard references of short-chain fatty acids (SCFAs, butyric acid, propionic acid, and acetic acid) were obtained from Sigma Aldrich (Shanghai, China). All other chemical reagents used were purchased from Sinopharm (Shanghai, China).

### 2.2. Animal Experiment

#### 2.2.1. Experimental Animals and Treatment

Four-week-old specific pathogen free (SPF) Sprague–Dawley (SD, male) rats (100 ± 10 g) were provided by Beijing Vital River Laboratory Animal Technology Co., Ltd. (Beijing, China, SCXK (Jing) 2016-0006). The following animal living conditions and experimental procedures abided by rules in the Guide for the Care and Use of Laboratory Animals.

The rats were kept in a ventilated rack system (22–24 °C and 12 h light/dark cycles). After 1 week of acclimation to the housing conditions, the animals were randomized into the following four groups (according to the weight of rats and randomized block design), all of which contained rats with similar mean body weights (BWs, 177 ± 4 g): a normal chow diet control group (NC, *n* = 7), a high-fat diet alone group (HF, *n* = 7), a high-fat + low quinoa intake diet group (HF + LQ, *n* = 7), and a high-fat + high quinoa intake diet group (HF + HQ, *n* = 7). Proper sample size calculation is both a scientific and ethical imperative. In accordance with the 3R’s, studies should be designed to reduce the number of animals used to meet scientific objectives. Here, we designed the animal experiment according to previous study [18,19,20], considering unexpected death of the animal during the intervention.

The high fat diet was prepared by mixing the normal chow diet (53.65%) with lard (20%, wt/wt), sucrose (10%), casein (10%), maltodextrin (2.7%), premix compound (1.9%), cholesterol (1.25%), and cholate (0.5%). The low quinoa intake feed (HF + LQ) included 9% quinoa (equivalent to 100 g/d of human intake), and the high quinoa intake feed (HF + HQ) included 27% quinoa (equivalent to 300 g/d of human intake). The feed components of two additive amounts of quinoa were as follows: the normal diet was 44.65% and 26.65%, the additive amount of quinoa was 9% and 27%, respectively. The weight ratios of other components were the same as those of the HF diet. Nutrients content of quinoa used in the present study was determined and shown in Table 1. The nutrients composition of feed and corresponding feed calories was described in our previous study [21]. The special feed (high-fat diet and quinoa diet) was manufactured by FBSH Biotechnology Co., Ltd. (Shanghai, China).

All rats were observed daily and kept healthy, and the body weights (BWs) of all animals were weighed once a week, water and food were made available ad libitum daily. Fecal samples were collected from all rats every morning at the beginning, 6th week, and 12th week of the experiment and flash frozen in liquid nitrogen immediately.

After 12 weeks of intervention, the rats were fasted for 12 h, weighed, and sacrificed after anesthetization (2% sodium pentobarbital, 0.2 mL/100 g). Blood samples were drawn from the abdominal aorta after sacrifice, and serum was obtained by low-speed centrifugation (3509× *g*, 10 min, 4 °C). Other samples, such as liver, spleen, and perirenal adipose tissue, were collected immediately and weighed; liver tissue was separated from the liver lobe for fixation of pathological sections; and then all the left sample was flash frozen in liquid nitrogen for subsequent analyses.

#### 2.2.2. Determination of the Liver Index and Spleen Index

The rats were weighed before sacrifice. The spleen and liver were carefully removed after sacrifice, washed with sterilized normal saline, and the excess water was absorbed with clean filter paper. Then, the mass of organs was weighed accurately using an analytical balance. The organ index was calculated according to the following formula: organ index = [weight _organ_ (g)/weight _body_ (g)] × 100% [22].

#### 2.2.3. Histological Analysis

Liver samples were fixed in 4% paraformaldehyde immediately after separation from the liver lobe, then, processed and embedded in paraffin for hematoxylin-eosin (HE) staining. The pathological changes were observed and reordered using Image and Microsuite (Olympus Soft Imaging Solutions GmbH, Muenster, Germany).

#### 2.2.4. Quantification of Hepatic Lipid

The content of hepatic lipid was measured according to our previous method [23]. Briefly, chloroform/methanol (20 mL, 2:1, *v/v*) was added to the liver tissue (1 g), the mixture was homogenized and sonicated at 400 W for 20 min. Then, the mixture was centrifuged and the supernatant was collected. Next, 0.2 times the volume of normal saline was added to the supernatant, mixed, and the lower liquid layer was collected after centrifugation. The sample was oven heated to a constant weight and the total mass of lipid in 1 g of liver tissue was assessed by weighing.

#### 2.2.5. Determination of NEFA, TG and TC Contents in Liver or Perirenal Adipose Tissues

For the determination of NEFA content in liver tissue, 50 mg of liver tissue from each rat was extracted using a tenfold-fold volume of ethanol, and 4 μL of supernatant was pipetted for analysis after centrifugation.

For the determination of TG, TC, and NEFA in perirenal adipose tissue, 50 mg of tissue from each rat was extracted using a tenfold volume of ethanol; and 2.5 μL of supernatant was pipetted for TG analysis (required 5 times dilution) and TC analysis (without dilution), respectively, and 4 μL of supernatant was pipetted for NEFA analysis.

The colorimetric reaction was performed following the manufacturer’s instructions.

#### 2.2.6. Analysis of Levels of Hepatic Antioxidative Parameters and Cytokines

The activities of GSH-PX and SOD and the levels of MDA and GSH in liver tissue were measured according to the instructions of commercial kits. Briefly, 100 mg of liver tissue from each rat was extracted and homogenized using a tenfold-fold volume of normal saline. After centrifugation, the supernatant volume was pipetted for the analysis of SOD activity (50 μL, 160-fold dilution), GSH-PX activity (200 μL, 20-fold dilution), GSH content (100 μL), and MDA content (50 μL). Colorimetric reactions were performed following the manufacturer’s instructions.

The levels of TGF-β, TNF-α, IL-10, and leptin in serum were measured according to the commercial ELISA kit. The serum was diluted two times for the determination of the leptin content.

#### 2.2.7. Determination of Liver Function Parameters

Liver function indexes, including serum albumin (ALB), alkaline phosphatase (ALP), alanine transaminase (ALT), aspartate aminotransferase (AST), total protein (TP), and globulin (glob), were analyzed using a Hitachi 7600 analyzer (Hitachi Ltd., Tokyo, Japan).

#### 2.2.8. Gut Microbiome Analysis Using 16S Ribosomal RNA (rRNA) Gene Sequencing

Gut microbiome analysis was conducted by the Medical Laboratory of Nantong ZhongKe (Jiangsu China). Briefly, the genomic DNA of fecal bacteria was extracted using a QIAamp DNA stool mini kit (Qiagen, Germany). The quality of the extracted DNA was evaluated via quantification analysis (NanoDrop ND-1000 spectrophotometer, NanodropTechnologies, Wilmington, DE, USA) and agarose gel electrophoresis. The V3-V4 hypervariable regions of the 16S rRNA genes were amplified with polymerase chain reaction (PCR), which were carried out with 15 μL of Phusion^®^ High-Fidelity PCR Master Mix (New England Biolabs), 2 μM of forward and reverse primers (forward primer, 5′-ACTCCTACGGGAGGCAGCA-3′ and reverse primer, 5′-GGACTACHVGGGTWTCTAAT-3′), and approximately 10 ng of template DNA. Thermal cycling parameters were set up as follows: initial denaturation at 98 °C for 1 min, followed by 30 cycles of denaturation at 98 °C for 10 s, annealing at 50 °C for 30 s, and elongation at 72 °C for 30 s. Finally, the samples were incubated at 72 °C for 5 min. PCR products were purified by using an AxyPrep DNA Gel Extraction Kit (Axygen, Axygen Biosciences, Union City, CA, USA) and a DNA library was constructed using a TruSeq Nano DNA LT Library Prep Kit (FC-121-4001; Illumina, San Diego, CA, USA). A Quant-iT PicoGreen dsDNA Assay Kit was used to quantify the library on a Promega QuantiFluor System. The optimized library was tested using an Agilent High Sensitivity DNA Kit (5067-4626; Agilent, Santa Clara, CA, USA) and subsequently sequenced on the Illumina Nova platform [23]. Bioinformatics analysis was performed by Passennor Biological Technology Co., Ltd. (Shanghai, China).

#### 2.2.9. Analysis of SCFAs Using Gas Chromatography-Mass Spectrometer (GC–MS)

Briefly, 0.6 mL of ultra-pure water was added to 0.2 g of fecal sample and ground until homogenous; 0.4 mL of the supernatant was pipetted after centrifugation (12,000 rpm for 20 min), and H_2_SO_4_ (0.1 mL, 50%, *v/v*) and ether solution (0.5 mL, containing the internal standard 2-methylpentanoic acid at 20 μg/mL) were added and vortexed until evenly distributed. Then, the mixture was placed in a refrigerator (4 °C) for 30 min after centrifugation (12,000 rpm for 10 min). The upper ether layer was pipetted for analysis.

The gas chromatography-triple quadrupole mass spectrometry (GC–MS, Agilent 7890B-7000D, Palo Alto, CA, USA) analytical conditions were as follows: carrier gas, helium (99.999%); flow rate, 1 mL/min; DB-FFAP column (30 m × 0.25 mm × 0.25 μm, Agilent Technologies); injection volume, 1.00 μL; injector temperature, 260 °C; split ratio, 5:1; column temperature, 100 °C maintained for 1 min then raised to 145 °C at 5 °C/min and to 240 °C at 15 °C/min and held for 6 min; interface temperature, 260 °C; ion source temperature, 230 °C; quadrupole temperature, 150 °C; and ionization mode, positive electron impact (EI+) at 70 eV. The internal standard curve method was used for quantification.

### 2.3. Statistical Analysis

GraphPad Prism 8.0 (GraphPad Software, San Diego, CA, USA) was used to visualize the parameters. Differences among groups were analyzed using one-way analysis of variance (ANOVA) followed by the least significant difference (LSD) multiple comparison test, and *p* value of <0.05 was considered statistically significant (IBM SPSS Statistics 25, IBM Co., Armonk, NY, USA).

## 3. Results

### 3.1. The Effect of Different Quinoa Intake Levels on the Food Intake and Weight of Rats Fed a High-Fat Diet

The food intake and the change of body weight are shown in Figure 1A,B. Overall, the average food intake of the rats in the HF group was (21.3 ± 2.6) g, significantly lower than that of the rats in the NC group (25.2 ± 2.4) g (*p* < 0.01); the food intake of two quinoa intake groups were (22.4 ± 1.9) g (*p* < 0.01) and (21.8 ± 2.6) g, respectively, higher than that of the rats in the HF group. From the second week of the experiment, the BW of rats in the HF group was higher than that of rats in the NC and two quinoa treatment groups.

### 3.2. The Effect of Different Quinoa Intake Levels on the Liver Index and Spleen Index of Rats Fed a High-Fat Diet

A long-term high-fat diet leads to lipid metabolism disorders, which results in an abnormal liver morphology, TG accumulation, and eventually causes hepatic steatosis. In the present study, the high-fat diet significantly increased the liver index compared with that of the NC group (*p* < 0.005), which was 2.62, 5.54, 5.10, and 4.51 for the NC, HF, HF + LQ, and HF + HQ groups, respectively. Quinoa intake, especially high quinoa intake, significantly inhibited abnormal liver expansion compared with the HF group (*p* < 0.005) (Figure 1C).

Splenomegaly is a common feature of patients with advanced chronic liver disease, such as cirrhosis [24]. The association between spleen dimension and NAFLD has been evaluated in a previous Japanese study [25]. In the present study, we found that 12 weeks of high-fat diet feeding led to a significant increase in the spleen index, which increased by 81.25% compared with that of rats in the NC group (*p* < 0.005), while it was close to the level of the NC group in rats in the high-intake quinoa group (Figure 1D).

### 3.3. The Effect of Different Quinoa Intake Levels on the Lipid Content in Liver and Adipose Tissue and Pathological Changes in Rats Fed High-Fat Diet

Lipids content analysis showed that a high-fat diet alone significantly induced the accumulation of fat and the content of free fatty acids (*p* < 0.01) in liver tissue, while the contents of lipids and NEFAs in rats of the two quinoa treatment groups were obviously lower than those of rats in the high fat diet alone group (*p* < 0.01) (Figure 1E,F). Moreover, the levels of TC (*p* < 0.005) and NEFA (*p* < 0.005) and the level of TG in perirenal adipose tissue of rats in the HF group increased compared with those of NC group; while the contents of TC and NEFA in the rats of the two quinoa treatment groups significantly decreased by 59.37% and 61.52% and 32.57% and 24.11% (*p* < 0.005), respectively, compared with those of rats in the HF group and showed no significant difference compared with those of rats in the NC group (*p* > 0.05) (as shown in Figure 1G–I).

Consistently, HE staining for pathological detection further verified the above analysis results. As shown in Figure 1J, the hepatocytes of rats in the NC group were intact without fat accumulation, and hepatic lobules were clear; while in the hepatic tissue of rats in the HF group, there were large lipid droplets, balloon-like degeneration, multiple inflammatory aggregations of hepatocytes. In the rats of the two quinoa treatment groups, hepatic intracellular lipid drop accumulation was alleviated compared with that of rats in the HF group.

### 3.4. The Effect of Different Quinoa Intake Levels on Hepatic Antioxidative Parameters and Serum Cytokines of Rats Fed a High-Fat Diet

It was shown that 12 weeks of a high-fat diet alone led to a significant decrease in hepatic antioxidative enzyme activities (SOD and GSH-PX) (*p* < 0.01) and hepatic GSH levels (*p* < 0.01) and an increase in hepatic MDA levels compared with those of rats in the NC group. Compared with the HF group, quinoa treatment effectively rescued the decrease in the activities of SOD (*p* < 0.01) and GSH-PX (*p* < 0.01) and the increase in MDA (*p* < 0.01). In addition, the hepatic GSH level of rats in the HF + LQ group was significantly higher than those of rats in the HF + HQ and HF groups (*p* < 0.05) (Figure 2A).

Oxidative stress is usually accompanied by inflammatory reactions. As shown in Figure 2B, a long-term high-fat diet caused chronic inflammation in rats. A high-fat diet alone significantly increased the serum concentrations of TNF-α (*p* < 0.01) and IL-10 (*p* < 0.05), and slightly increased the concentration of TGF-β and decreased the level of leptin compared with the NC group. The levels of TNF-α, IL-10, TGF-β, and leptin in rats of HF+HQ group were close to those of the NC group (*p* > 0.05) (Figure 2B).

### 3.5. The Effect of Different Quinoa Intake Levels on Liver Function Indexes of Rats Fed a High-Fat Diet

There were no significant differences in the activities of ALT, ALP, the levels of TP and ALB among groups at the beginning of the experiment (t_0_) (*p* > 0.05) (as shown in Table 2). At the end of the experiment (t12), the high-fat diet alone obviously increased the levels of ALT (*p* < 0.01), ALP activities (*p* < 0.005), TP, and Glob (*p* < 0.005) and decreased the A/G ratio (*p* < 0.005); quinoa intervention, especially high quinoa intake, effectively ameliorated the liver injury induced by hepatic steatosis, which exhibited lower ALT activity (*p* < 0.05), lower levels of TP (*p* < 0.005) and Glob (*p* < 0.01), and a higher A/G ratio than those of rats in the HF group (*p* < 0.05) (Table 2).

### 3.6. The Effect of Different Quinoa Intake Levels on the Gut Microbiota Diversity of Rats Fed a High-Fat Diet

Diet is one of the environmental factors that deeply influences the gut microbiota. In our study, we analyzed the alpha diversity and beta diversity of the fecal gut microbiota. At the beginning of the experiment, the Chao1 index of the HF group was significantly higher than that of the NC group (*p* < 0.05), and there were no significant differences among the two quinoa treatment groups and the NC group. While a high-fat diet led to a decrease in the Chao1 index, that of the HF group (*p* < 0.005) and HF + LQ group (*p* < 0.05) was significantly lower than that of the NC group at the 6th week of the experiment, while there was no significant difference between the HF + HQ group and the NC group. At the 12th week, the Chao1 index of the high-fat diet treatment groups was lower than that of the NC group; however, that of the HF + HQ group was closer to that of the NC group (Figure 3A).

For the Shannon and Simpson indexes, there were no significant differences among the four groups at the beginning and the end of the experiment. However, quinoa demonstrated a better trend in maintaining in gut bacterial diversity; the Shannon and Simpson indexes of the HF + LQ and HF + HQ groups were higher than those of the NC and HF groups after 12 weeks of intervention, and high quinoa intake exhibited a more obvious effect (Figure 3A).

Principal coordinates analysis (PCoA) plots of beta-diversity were constructed using the Bray Curtis Distance and analysis of differences between groups at the beginning, the 6th week, and the end of the experiment. PCoA analysis reflects similarity or difference in the sample community composition. The structure and composition of the gut microbiome differed among the four groups (Figure 3B).

### 3.7. The Effect of Different Quinoa Intake Levels on the Gut Microbiota Distribution of Rats Fed a High-Fat Diet at the Phylum Level

The intestinal microbiota of the experimental rats was mainly divided into six phyla, among which *Firmicutes* and *Bacteroidetes* accounted for 60% and 15% of the total abundance, respectively; other species, such as *Verrucomicrobia* and *Proteobacteria*, accounted for a smaller proportion of the total abundance. In the present study, taxonomic compositional analysis showed that the relative abundances of the main bacterial microbiota at the phylum level fluctuated during the whole experimental period (Figure 4A). Therefore, it was more meaningful to analyze the ratio between *Firmicutes* and *Bacteroidetes* (F/B) abundances, which were 4.8 ± 4.6, 1.9 ± 0.9, 0.9 ± 0.3, and 1.7 ± 0.9 for the NC, HF + LQ, HF + HQ, and HF groups, respectively, at the beginning of the experiment. The F/B of the three high-fat diet treatment groups was significantly lower than that of the NC group (*p* < 0.05). The F/B of the above four groups were 3.3 ± 2.3, 2.3 ± 2.4, 1.0 ± 0.3 (*p* < 0.05 vs. NC group), and 2.1 ± 0.9, respectively, at the 6th week and were 4.4 ± 0.8, 5.9 ± 4.1 (*p* < 0.05 vs. t_0_ and t_6_), 14.0 ± 18.8 (*p* < 0.05 vs. t_0_ and t_6_), and 17.0 ± 23.0 (*p* < 0.05 vs. t_0_), respectively, at the 12th week. It was obvious that a high-fat diet promoted an increase in the F/B; however, quinoa intake could alleviate the dysbiosis between *Firmicutes* and *Bacteroidetes* to a certain degree (Figure 4B).

*Proteobacteria* and *Verrucomicrobia* are two important intestinal microbiota phyla, although they account for a low ratio. In this study, we found that a high-fat diet increased the relative abundance of *Proteobacteria* (Figure 4C). The relative abundances of *Proteobacteria* were 3.0 ± 1.8%, 0.8 ± 0.5%, 0.8 ± 1.3%, and 3.2 ± 3.7% for the NC, HF + LQ, HF + HQ, and HF groups, respectively, and no significant difference was observed among the four groups at the beginning of the experiment (*p* > 0.05). At the 6th week of intervention, the relative abundances of *Proteobacteria* were 0.8 ± 1.2%, 12.4 ± 12.0%, 7.7 ± 4.1%, and 11.6 ± 5.2% for the NC, HF + LQ (*p* < 0.01, vs. NC group), HF + HQ, and HF (*p* < 0.01, vs. NC group) groups, respectively. At the end of the experiment, the relative abundances of *Proteobacteria* were 0.5 ± 0.4%, 10.0 ± 10.0%, 10.5 ± 5.9%, and 12.0 ± 8.8% for the NC, HF + LQ (*p* < 0.05, vs. NC group), HF + HQ (*p* < 0.05, vs. NC group), and HF groups (*p* < 0.01, vs. NC group), respectively. It was shown that quinoa could inhibit the overgrowth of *Proteobacteria*.

The relative abundance of *Verrucomicrobia* in the NC, HF + LQ, and HF groups significantly decreased compared with those at the beginning of the experiment (*p* < 0.05), but the reduction in the HF group was higher than that in the HF + LQ group. However, the relative abundance of this species was significantly increased in the HF + HQ treatment group at the 6th week and the end of the experiment (*p* < 0.01) and was higher than that in the NC (*p* < 0.01), HF + LQ (*p* < 0.05), and HF (*p* < 0.05) groups at the end of the 6th week as well as the end of the experiment (*p* < 0.01) (Figure 4D).

### 3.8. The Effect of Different Quinoa Intake Levels on the Gut Microbiota Distribution of Rats Fed a High-Fat Diet at the Genus Level

We further analyzed the taxonomic composition of the intestinal microbiota at the genus level. It was shown that intestinal microbiota bacteria, such as *Lactobacillus*, *Blautia*, *Bacteroides*, *Dorea*, and others, fluctuated form the beginning to the end of the experiment (Figure 5A).

Principal component analysis (PCA) can quantify the degree of variation in species composition between groups. The gut microbial composition of the rats belonging to the NC group was markedly separated from those of the high-fat diet treatment groups after 6 and 12 weeks of intervention, whereas quinoa intervention groups, especially HF + HQ groups, were distinct from the HF group, although there were some individual differences (Figure 5B).

Linear discriminant analysis (LDA) effect size (LEfSe) results exhibited remarkably enriched species in each group. Briefly, a high-fat diet alone induced enrichment of *Shigella*, while high quinoa intake promoted the enrichment of *Ruminococcus*, *Akkermansia*, and *Parabacteroides* and low quinoa intake promoted the enrichment of *Blautia* and *Dorea* at the genus level (Figure 5C).

Heatmap cluster analysis clearly revealed the main bacterial abundance comparison among the four groups (Figure 5D). Specifically, a high-fat diet alone led to an increase in the relative abundances of *Blautia* (*p* < 0.01), *Ruminococcus*, *Dorea* (*p* < 0.01), *Shigella* (*p* < 0.05), and *Clostridium* and a decrease in the abundance of *Lactobacillus* (*p* < 0.01) compared with that of the NC group at 6th and the end of the experiment. Quinoa intake did not reverse the decrease in *Lactobacillus* compared with the HF group. However, high quinoa intake could significantly increase the relative abundance of *Akkermansia* (*p* < 0.05), and lower quinoa intake could significantly increase the relative abundance of *Blautia* at the 6th week and the 12th week compared with the NC and HF groups (*p* < 0.05). Notably, the relative abundances of *Blautia* and *Dorea* of rats in the HF + HQ group were lower than those of rats in the HF + LQ group at the end of the experiment; the relative abundances of *Clostridium* and *Turicibacter* of rats in both quinoa intervention groups were significantly decreased at the 6th week of the experiment (*p* < 0.05) and were lower than those of rats in the HF group at the end of the experiment (*p* < 0.05) (Figure 5E).

The correlation analysis between the abundances of specific gut bacterial and transaminase activity, hepatic lipid and NEFA contents is shown in Figure 5F. The relative abundance of *Lactobacillus* was negatively related to the levels of hepatic lipids and NEFAs, while the relative abundances of *Blautia*, *Shigella*, *Dorea*, and *Clostridium* were positively related to the levels of hepatic lipids and NEFAs. In addition, the abundances of *Shigella* and *Dorea* were positively related to the activity of ALT.

### 3.9. The Effect of Different Quinoa Intake Levels on SCFAs

SCFAs are one of the important metabolites of gut bacteria when they consume food rich in dietary fiber. In the present study, the levels of acetic acid, propionic acid, and butyric acid in the quinoa intake group were slightly higher than those in the HF group. The level of total SCFA in the HF group was significantly lower than that in the NC group (*p* < 0.05). However, there were no significant differences between the two quinoa intervention groups and the NC group (Figure 6).

## 4. Discussion

NAFLD is considered as the hepatic manifestation of metabolic syndrome. It exhibits the inflammatory reaction, oxidative stress in hepatic tissue and gut microbiota dysbiosis and will progression to NASH without intervention strategy [26].

Clinical data showed that NAFLD patients disclose a relatively high prevalence of hepatomegaly and splenomegaly, a common feature of patients with advanced chronic liver disease, such as viral hepatitis and cirrhosis, which is probably due to hemodynamic alteration of portal flow toward hypertension, as a consequence of liver fibrosis [24]. In animal experiment, the organ index can be used to evaluate the overall organ status. An abnormal increase of the parenchymatous organ index is usually associated with some damage to the organ, which exhibits swell or accrete [22]. In this study, we found that quinoa treatment could effectively inhibit the abnormal increase of liver and spleen indexes in NAFLD rats. This may attribute to its beneficial role in the prevention of NAFLD, such as ameliorating hepatic adiposopathy, rescuing inflammatory reactions and oxidative stress, and regulation of the gut microbiota dysbiosis.

First, we found that a high-fat diet led to an obvious increase in NEFA in both liver and adipose tissue. Quinoa intake significantly reduced the levels of NEFA in liver and adipose tissue and alleviated hepatic steatosis. Pathological results also revealed the improvement effect of quinoa on hepatic steatosis. NEFAs are one of the primary sources of free fatty acids (FFAs) from which hepatic lipids accumulate. Excessive FFAs in the liver are esterified to form TG, which accumulates in the liver and causes steatosis. FFAs themselves have strong cytotoxicity; they can induce endoplasmic reticulum stress, increase lysosomal permeability, lead to mitochondrial dysfunction, change the gene expression of cytolethal factors, cause liver cell necrosis, and lead to liver lesions [27]. Elevated NFFA levels also cause IR in the liver, which contributes to the pathogenesis of type two diabetes mellitus (T2DM) and causes low-grade inflammation as well as the development of NAFLD [28]. Adipose tissue-derived NEFA stimulates hepatic dendritic cells (DCs) and monocyte/macrophage accumulation, thus recapitulating the pathology of the fatty liver [29]. Serious hepatic steatosis causes NASH, which further leads to abnormal liver function, especially increased transaminase activities. In the present study, it was shown that 12 weeks of high-fat diet consumption significantly increased the levels of ALT, ALP activities, and TP and decreased the A/G ratio. Quinoa intervention, especially a high amount of quinoa intake, lowered ALT activity. These findings suggest the health attributes of quinoa in regulating lipid metabolism disturbances, reducing hepatic steatosis and preventive effect on the progression of NAFLD due to the comprehensive effect of functional components, such as flavonoids, polyphenols, saponins, and other nutrients [30].

Second, a long term of high-fat diet induces obesity or overweight which further promotes the generation of excessive reactive oxygen species (ROS) and chronic inflammation reaction. The increased generation of ROS changes the insulin sensitivity, affects the expression and activity of key enzymes involved in lipid metabolism. Further, the interaction between redox signaling and innate immune signaling forms a complex network that regulates inflammatory responses [31]. In the present study, it was shown that 12 weeks of high-fat diet feeding significantly decreased hepatic antioxidative enzyme (SOD and GSH-PX) activities and hepatic GSH levels and increased the concentrations of serum TNF-α and IL-10. Previous studies also found increased levels of TNF-α and IL-10 in serum or liver tissue in obesity-related NAFLD rat and mouse models established by high-fat diet or monosodium glutamate (MSG) injection [32,33]. TNF-α and IL-10 play a bridging role in IR in NAFLD. It has been found that a long-term high-fat diet can trigger the transformation of liver Kupffer cells (KC) into the M1 phenotype, resulting in the up-regulated expression of pro-inflammatory cytokines, such as TNF-α, which could further aggravate fat accumulation in hepatocyte, thus affecting the fatty acid oxidation, TG accumulation, and IR of hepatocytes [34,35]. Fujisaka et al. [36] found that the number of M1-type macrophages and M1/M2 are closely related to the pathogenesis and development of IR. A high-fat diet can induce adipose tissue macrophages (ATMs) to polarize towards M1-type macrophages. The anti-inflammatory Th2 cytokine IL-10 can increase the number of M2-type macrophages, lead to congenital immune imbalance, and eventually lead to hepatocyte steatosis. In present study, we found that quinoa treatment effectively rescued the oxidative stress and chronic inflammation reaction induced by a long-term high-fat diet. These health benefits were attributed to the antioxidative and anti-inflammation properties of quinoa. Our study also indirectly verified the evidence implicating quinoa as an important contributor to immune nutritional health [10,21,37].

In recent years, the importance of the gut–liver axis has drawn attention in the occurrence and development of some chronic diseases [38]. In this study, we investigated the influence of quinoa intake on gut microbiota dysbiosis in NAFLD rats. It was shown that a high-fat diet led to a decrease in colon microbial richness and diversity after 6 weeks of the experiment, which is consistent with a previous study [39]; while a high amount of quinoa could more effectively rescue the decrease in the richness and diversity of colon bacteria. However, a recent study assessed the efficacy of quinoa with saponin and quinoa without saponin on obesity and showed that a high-fat diet induced an increase in the microbial richness and diversity of colon bacteria, but quinoa treatment did not effectively influence the bacterial diversity, which is not consistent with the results of our study [40]. This may be due to the difference in intervention time and quinoa intakes, which implied that the diversity of intestinal flora may be effectively regulated only when a certain amount of quinoa or specific diet is consumed over a period of time.

*Firmicutes* and *Bacteroidetes* are the two most important bacterial phyla in the gastrointestinal tract. The F/B ratio is widely accepted to have an important influence in maintaining normal intestinal homeostasis. An increased or decreased F/B ratio is regarded as dysbiosis, whereby the former is usually observed with obesity [41]. Besides, a high-fat diet usually leads to a decrease in some probiotics (such as *Lactobacillus* spp.) and the increase in pathogenic bacteria. In our study, it also showed that a high-fat diet led to an increased F/B and decrease in the abundance of *Lactobacillus*; however, quinoa intake could inhibit the dysbiosis between *Firmicutes* and *Bacteroidetes* to a certain degree. Though quinoa diet did not reverse the decrease in *Lactobacillus*, notably, a high intake of quinoa could significantly increase the relative abundance of *Akkermansia*, while a low intake of quinoa could significantly increase the relative abundance of *Blautia*. *Akkermansia muciniphila* is a promising candidate as a probiotic and can be used as a probiotic for the treatment of obesity, diabetes, and atherosclerosis [42,43,44]. Kim S et al. [45] demonstrated that *Akkermansia muciniphila* prevented fatty liver disease by regulating the expression of genes that regulate fat synthesis and inflammation in the liver. Interestingly, although *Blautia* is a producer of butyric acid, exogenous butyrate is an effective substance for attenuating NAFLD [46]; multivariate analyses indicated an increase in *Blautia* and *Dorea* abundances in NASH patients compared to those in healthy controls [47]. A new study showed that metabolic syndrome patients harbor a microbiome enriched in *Enterobacteriaceae*, *Turicibacter sp.*, *Clostridium coccoides*, and *Clostridium leptum*, while beneficial taxa, such as *Akkermansia muciniphila*, are downregulated [48]. The relative abundances of *Clostridium* and *Turicibacter* of rats in the two quinoa intervention groups were significantly decreased during the intervention time and were lower than those of rats in the HF group at the end of the experiment. Therefore, the regulation effect of quinoa on the gut microbiota suggests that quinoa may play a positive role in the daily diet management for the prevention of NAFLD occurrence, but the specific effective amount and intervention time for human being still need more exploration in clinical trials.

SCFAs are considered important for immune function and gut health [49,50]. As an intestinal nutrient, SCFAs can activate the differentiation and apoptosis of normal intestinal epithelial cells, promote cell regeneration to repair intestinal mucosa, stimulate the synthesis of intestinal mucins and glycoproteins, and enhance the protective function of the mucous layer [51]. In the present study, it was shown that a high-fat diet led to a significant decrease in SCFA, while quinoa intake could slightly rescue the decrease in SCFA. This result may suggest the beneficial effect of quinoa on NAFLD, but the effectiveness is limited because of individual difference.

## 5. Conclusions

In summary, our results demonstrated some beneficial effects of different quinoa intake (equivalent to 100 g or 300 g of daily human intake) on reducing the levels of NEFA in liver and adipose tissue, improving hepatic steatosis, oxidative stress and inflammation response under a high-fat diet. These beneficial effects may be attributed to its regulation effect on gut microbiota dysbiosis, including bacterial diversity, F/B ratio, and the relative abundances of *Akkermansia*, *Clostridium*, and *Turicibacter* at the genus level. Quinoa (especially high intakes of quinoa) may be suggested to be a cereal food selection for prevention of NAFLD in our daily life. However, the prevention capacity of quinoa for the progression of NAFLD was still limited, while the duration and quinoa intakes of intervention are still worthy of further study in clinical trial.

## Figures and Tables

**Figure 1 foods-12-01780-f001:**
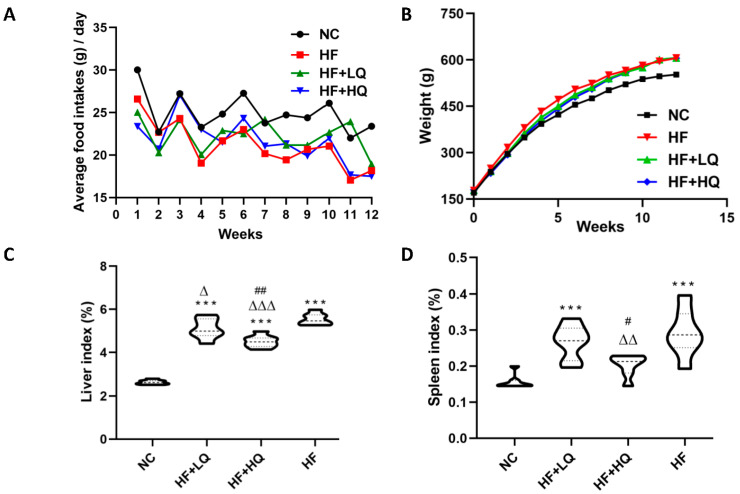
The effect of different quinoa intake levels on food intake, body weight lipid content in liver and perirenal adipose tissue, and pathological change of rats fed a high-fat diet. (**A**) Average food intake of rats of each day during the whole experiment; (**B**) The change of BW of rats during the whole experiment; (**C**) Liver index; (**D**) Spleen index; (**E**) The lipid content in liver tissue; (**F**) NEFAs content in liver tissue; (**G**–**I**) The contents of TG, TC and NEFAs in perirenal adipose tissue; (**J**) Hematoxylin-eosin (HE) staining of liver tissue from the rats in the different groups (magnification × 200). NC: normal control group (*n* = 7); HF: high-fat diet alone group (*n* = 7); HF + LQ: low quinoa intake group (*n* = 7); HF + HQ: high quinoa intake group (*n* = 7). ** *p* < 0.01, *** *p* < 0.005 vs. the NC group; ^Δ^
*p* < 0.05, ^ΔΔ^
*p* < 0.01, ^ΔΔΔ^
*p* < 0.005, vs. the HF group; ^#^
*p* < 0.05, ^##^
*p* < 0.01, vs. the HF + LQ group.

**Figure 2 foods-12-01780-f002:**
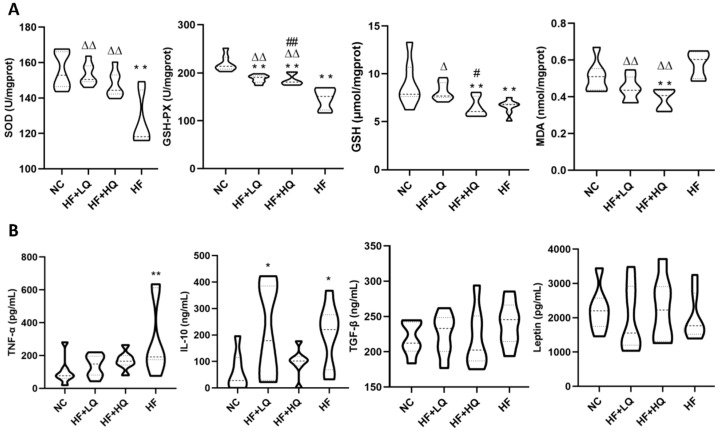
The effect of different quinoa intake levels on antioxidative parameters and cytokines of rats fed a high-fat diet. (**A**) Changes in hepatic antioxidative parameters. (**B**) Changes in serum cytokines and leptin. * *p* < 0.05, ** *p* < 0.01, vs. the NC group; ^Δ^
*p* < 0.05, ^ΔΔ^
*p* < 0.01, vs. the HF group; ^#^
*p* < 0.05, ^##^
*p* < 0.01, vs. the HF + LQ group.

**Figure 3 foods-12-01780-f003:**
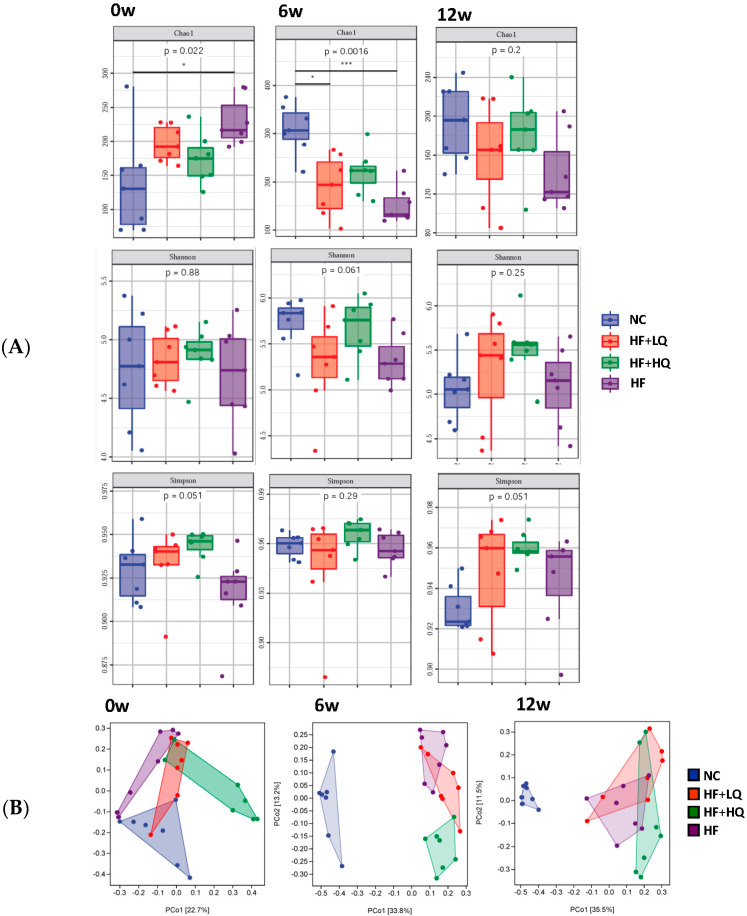
The effect of different quinoa intake levels on the gut microbiota diversity of rats fed a high-fat diet. (**A**) Alpha diversity (Chao1, Simpson, and Shannon index) of the gut microbiota in each group of rats at the beginning, 6th week, and end of the experiment; (**B**) PCoA plots of beta-diversity in each group at the beginning, 6th week, and end of the experiment. * *p* < 0.05; *** *p* < 0.005.

**Figure 4 foods-12-01780-f004:**
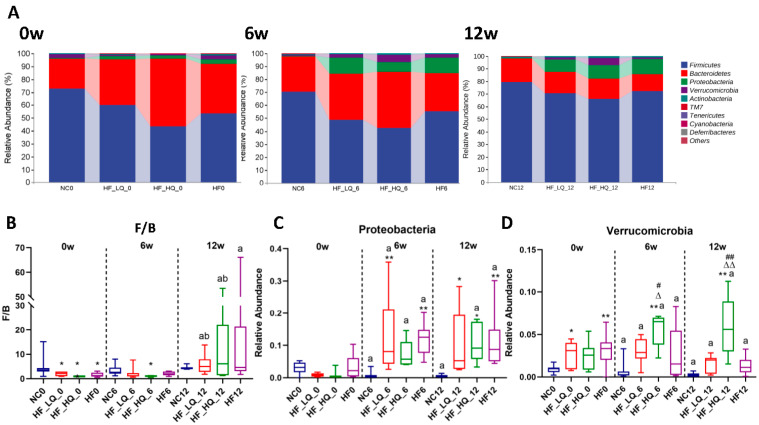
The effect of different quinoa intake levels on gut microbiota distribution of rats fed a high-fat diet at the phylum level. (**A**) Relative abundance of major microbial phyla in the different groups of rats at the beginning, 6th week, and end of the experiment; (**B**–**D**) Comparison of the ratio between *Firmicutes* and *Bacteroidetes* (F/B) and the relative abundances of *Verrucomicrobia* and *Proteobacteria* among the four groups. * *p* < 0.05, ** *p* < 0.01, vs. the NC group; ^Δ^
*p* < 0.05, ^ΔΔ^
*p* < 0.01, vs. the HF group; ^#^
*p* < 0.05, ^##^
*p* < 0.01, vs. the HF + LQ group. ^a^
*p* < 0.05, vs. the beginning of the experiment (t_0_); ^b^
*p* < 0.05, vs. the 6th week of the experiment (t_6_).

**Figure 5 foods-12-01780-f005:**
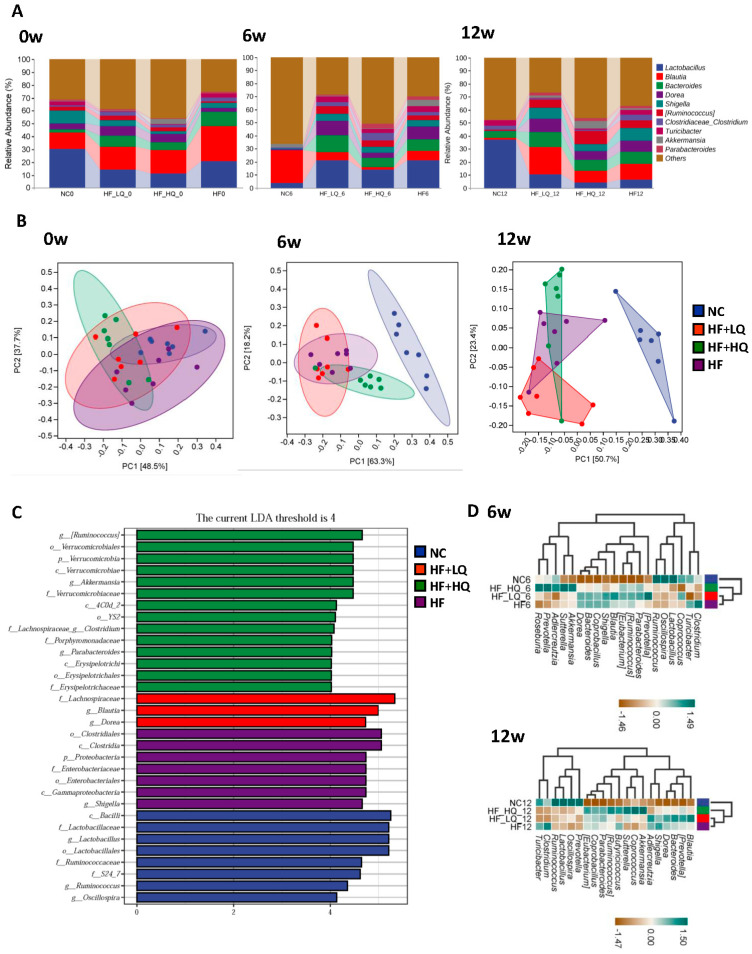
The effect of different quinoa intake levels on gut microbiota distribution of rats fed a high-fat diet at the genus level. (**A**) Relative abundance of major microbial genera in the different groups of rats at the beginning, 6th week, and end of the experiment. (**B**) PCA among groups at the genus level at the beginning, 6th week, and the end of the experiment. (**C**) Linear discriminant analysis (LDA) effect size (LEfSe) analyses at the 12th week; (**D**) Heatmap of species composition at the genus level at the 6th week and the 12th week; (**E**) The relative abundances of specific gut bacteria at the genus level. (**F**) The correlation analysis among the relative abundances of fecal bacteria and biochemical parameters. * *p* < 0.05, ** *p* < 0.01, vs. the NC group; ^Δ^
*p* < 0.05, ^ΔΔ^
*p* < 0.01, vs. the HF group; ^#^
*p* < 0.05, ^##^
*p* < 0.01, vs. the HF + LQ group. ^a^
*p* < 0.05, vs. the beginning of the experiment (t_0_); ^b^
*p* < 0.05, vs. the 6th week of the experiment (t_6_).

**Figure 6 foods-12-01780-f006:**
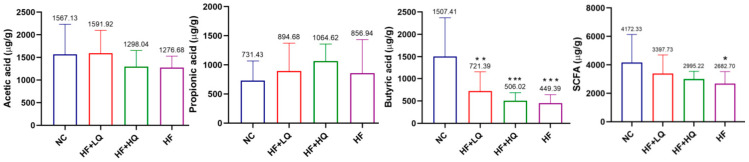
The effect of different quinoa intake levels on the level of SCFAs. * *p* < 0.05, ** *p* < 0.01, *** *p* < 0.001, vs. the NC group.

**Table 1 foods-12-01780-t001:** The contents of primary nutrients and bioactive components in quinoa used in the present study.

Primary Nutrients (g/100 g)		Elements (mg/kg)		Bioactive Components (g/100 g)	
Protein	13.34	K	6642	Flavonoids	0.528
Moisture	10.20	P	5303	Polyphenols	0.496
Ash	2.073	Ca	629	Saponins	0.723
Starch	56.52	Na	73.1		
Soluble dietary fiber	1.760	Fe	46.5		
Insoluble dietary fiber	7.912	Zn	28.0		
Fat	6.805	Mn	17.7		
		Cu	6.44		

**Table 2 foods-12-01780-t002:** Index of liver function in rats of each group.

Time	Indicator	NC	HF	HF + LQ	HF + HQ
t_0_	ALB(g/L)	44.43 ± 1.27	44.71 ± 0.76	45.29 ± 1.5	44.57 ± 0.53
	ALT(U/L)	53.43 ± 7.41	54.86 ± 5.18	52.86 ± 7.03	53.71 ± 5.22
	AST(U/L)	203.29 ± 19.47	255.29 ± 57.27 *	187.86 ± 15.32 ^ΔΔ^	206.71 ± 34.19 ^Δ^
	ALP(U/L)	376.43 ± 44.05	408.29 ± 37.14	386.86 ± 69.92	385.43 ± 68.13
	TP(g/L)	57.33 ± 2.29	59.11 ± 1.27	57.33 ± 2.06	59 ± 2.69
	Glob(g/L)	12.86 ± 0.9	14.29 ± 0.49 *	13.71 ± 1.11	12.71 ± 1.6 ^Δ^
	A/G	3.43 ± 0.21	3.16 ± 0.15	3.33 ± 0.2	3.54 ± 0.46 ^Δ^
t_12_	ALB(g/L)	39.86 ± 1.68	41.14 ± 1.77	41 ± 1.83	40.14 ± 0.69
	ALT(U/L)	45.00 ± 8.81	73.43 ± 27.26 **	63.86 ± 21.62	52.29 ± 12.26 ^Δ^
	AST(U/L)	146.14 ± 24.04	170.71 ± 39.8	174.43 ± 48.1	135.43 ± 4.79 ^#^
	ALP(U/L)	109.29 ± 16.87	168.43 ± 16.51 ***	159.43 ± 14.73 ***	156.57 ± 18.06 ***
	TP(g/L)	53.86 ± 1.68	62.14 ± 2.19 ***	61.43 ± 2.15 ***	57.43 ± 2.64 **^ΔΔΔ##^
	Glob(g/L)	14.14 ± 0.9	21.14 ± 0.9 ***	20.71 ± 2.36 ***	17.57 ± 2.88 **^ΔΔ##^
	A/G	2.8 ± 0.19	1.97 ± 0.08 ***	2.01 ± 0.26 ***	2.34 ± 0.48 **^Δ#^

* *p* < 0.05 vs. the NC group, ** *p* < 0.01 vs. the NC group; *** *p* < 0.005 vs. the NC group; ^Δ^
*p* < 0.05 vs. the HF group,^ΔΔ^
*p* < 0.01 vs. the HF group,^ΔΔΔ^
*p* < 0.01 vs. the HF group; ^#^
*p* < 0.05 vs. the HF + LQ group; ^##^
*p* < 0.01 vs. the HF + LQ group.

## Data Availability

The data presented in this study are available on request from the corresponding author.

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
