# Peer review of "Quinoa Ameliorates Hepatic Steatosis, Oxidative Stress, Inflammation and Regulates the Gut Microbiota in Nonalcoholic Fatty Liver Disease Rats"

_foods, 2023, doi:10.3390/foods12091780_

Round 1
Reviewer 1 Report
Quinoa ameliorates hepatic steatosis, oxidative stress, inflammation and regulates the gut microbiota in NAFLD rats
Thank you for the opportunity to review the above-titled manuscript. After going through the document, I have some issues with the content of the study. See my comments below:
1. Two groups are missing, that is normal rats treated with the high dose of the quinoa and a group treated with standard drug.
2. Is Table 1 an original data by the authors? If not give the reference.
3. Why GC–MS was used not NMR?
4. State the magnification for Figure 1.
Author Response
- Two groups are missing, that is normal rats treated with the high dose of the quinoa and a group treated with standard drug.
Answer: Thank you very much for the professional comments.
Firstly, no drug has been approved by the United States Food and Drug Administration or the European Medicines Agency for the treatments of NAFLD at present, and the primary strategies for the vast majority of NAFLD patients focuses on weight control, physical exercise and diet management【Journal of hepatology, 2017, 67(4): 829-846.】【Endocrinol Diabetes Metab, 2019, 3, e00105-e00105】【Gastroenterology & hepatology, 2021, 18, 633 373-392.】. Also, referring to previous NAFLD-intervention related studies 【Foods (Basel, Switzerland), 2021, 10(12): doi: 10.3390/foods10123022.】【Journal of agricultural and food chemistry, 2022, 70(46): 14732-14743.】, we did not set a positive drug control group.
Secondly, quinoa is one of the traditional staple foods of local people in South America Bolivia, such as Ecuador, Peru and Chile, and quinoa is safe food. According to the rules of animal protection, the use of animals in scientific research activities should be minimized. Therefore, we did not set the group of normal rats treated with quinoa.
Thirdly, the purpose of the present study was to investigate the beneficial effect of quinoa on rats fed a high fat diet, we compared the differences of rats between high fat diet treatment alone group and high fat diet + different amount of quinoa treatment group. We can obtain some meaningful data in present study.
- Is Table 1 an original data by the authors? If not give the reference.
Answer: Thank you very much for the professional question.
The data shown in Table 1 were determined by ourself. It was noted in revised manuscript, which was marked with red fonts, please refer to line 114 to 115.
- Why GC–MS was used not NMR?
Answer: Thank you very much for the professional question.
Yes, exactly, NMR can also be used to detect SCFA. NMR can effectively analyze compounds that are difficult to ionize, require derivatization, or have very high concentrations in comparison with GC-MS. But the main disadvantages of NMR are poor sensitivity and separation ability【TrAC Trends in Analytical Chemistry, 2022, 148: 116540.】.The composition of fecal metabolites is complex and some of short chain fatty acids (SCFAs) content in feces is low, therefore, the low separation ability and sensitivity of NMR limit its application in SCFAs analysis. GC-MS is a commonly used method for the detection of fecal SCFA in many published papers 【Phytomedicine : international journal of phytotherapy and phytopharmacology, 2020, 77: 153268.】【Foods (Basel, Switzerland), 2021, 10(4): doi: 10.3390/foods10040902.】. Hence, we use GC–MS in present study.
- State the magnification for Figure 1.
Answer: Thank you very much for the professional comment.
Magnification for Figure 1 is stated in figure legend, please refer to line 267, which was marked with red fonts in revised manuscript.
Reviewer 2 Report
The article corresponding to the ref. foods-2196153 Quinoa ameliorates hepatic steatosis, oxidative stress, inflammation and regulates the gut microbiota in NAFLD rats, it is aimed at uncovering the effects of quinoa on the risk factors associated with non-alcoholic fatty liver disease. For these, the authors implemented various analytical techniques to assess experimental animals on intestinal pathology, oxidative stress, inflammation, and dysbiosis. The analytical methods have been properly applied and the results, in general, correctly identified, providing valuable information about the research area that also fits well with the scope of the journal. However, some issues need to be addressed.
As a general comment, the authors should avoid the use of abbreviations in the title (NAFLD) that may be confusing for readers. This should be replaced by the complete denomination of the pathophysiological situation focusing the attention of the paper. Also, the abbreviations in the main text should be defined in the abstract, as well as the first time that this is used in additional sections in the article.
Concerning the experimental design, the "n" of the study is not enough justified. This is closely dependent on factors that the authors do not refer to in the text, such as the variability of the markers or the statistic test used to identify quantitative differences between research groups. So, additional information supporting the selection of the "n" should be provided or, alternatively, the requirements of the statistic test. For many markers, this "n" is not appropriate when determining in vivo research.
Also, the relevance of inflammation markers has been no addressed. The authors include both anti- and pro-inflammatory molecules and do not discuss appropriately their interest and the meaning of the findings. In this regard, the use of Il-10 in hepatocytes is especially hard to understand because this interleukin is almost specifically secreted by immune cells and not for parenchymal liver units. So, this fact would be responsible for the dispersion of the results (being almost within the limit of quantification of the method (normally in the picograms range), the interpretation of data relative to IL-10 should be reinterpreted and redrafted carefully.
Some subsections regarding methodology show a lack of proper citations while not enough details for reproduction are provided. This is the case of subsection 2.2.2. Determination of the liver index and spleen index. Beyond this, in this section, the formula included is misexpressed and should be correctly applied to both the liver and spleen determinations.
Additional information (description and discussion) on the organ index is required.
Concerning the modulation of microbiota, the description should be carefully reviewed. Especially for stressing adequately those effects of relevance. In this regard, almost all treatments and target species monitored produced no such desirable effects being closer to the High-fat group than the Control group. In this sense, the statements should be modulated indicating the limited capacity of quinoa to act on this risk factor for NAFLD, also, this information should be included in the conclusions section.
Author Response
Point-by-point responses to the Reviewers' comments
Reviewers' comments:
The article corresponding to the ref. foods-2196153 Quinoa ameliorates hepatic steatosis, oxidative stress, inflammation and regulates the gut microbiota in NAFLD rats, it is aimed at uncovering the effects of quinoa on the risk factors associated with non-alcoholic fatty liver disease. For these, the authors implemented various analytical techniques to assess experimental animals on intestinal pathology, oxidative stress, inflammation, and dysbiosis. The analytical methods have been properly applied and the results, in general, correctly identified, providing valuable information about the research area that also fits well with the scope of the journal. However, some issues need to be addressed.
As a general comment, the authors should avoid the use of abbreviations in the title (NAFLD) that may be confusing for readers. This should be replaced by the complete denomination of the pathophysiological situation focusing the attention of the paper. Also, the abbreviations in the main text should be defined in the abstract, as well as the first time that this is used in additional sections in the article.
Answer: Thank you very much for the professional comments.
We have carefully checked the whole manuscript, and made the correction in revised manuscript, which were all marked with red fonts.
Concerning the experimental design, the "n" of the study is not enough justified. This is closely dependent on factors that the authors do not refer to in the text, such as the variability of the markers or the statistic test used to identify quantitative differences between research groups. So, additional information supporting the selection of the "n" should be provided or, alternatively, the requirements of the statistic test. For many markers, this "n" is not appropriate when determining in vivo research.
Answer: Thank you very much for the professional comments.
Yes, the comments are quite right.
But our study is an exploratory study, it’s hard to obtain rational reference data to calculate the sample size according to statistic requirement. In animal experiment, the intervention condition is relatively easier to be controlled compared with clinical trial. Therefore,the effect size is relatively larger than clinical study,and required sample size in animal study will be more smaller. Therefore, we use same animal numbers with some previous studies 【J Immunol Res . 2021 Aug 17;2021:2264737. 】【Arch Biochem Biophys . 2021 Jul 15;705:108894.】【 Diagnostics (Basel), 2019. 9(4)】. We supplemented the reference in revised manuscript. Please refer to line 105, which was marked with red font
Also, the relevance of inflammation markers has been no addressed. The authors include both anti- and pro-inflammatory molecules and do not discuss appropriately their interest and the meaning of the findings. In this regard, the use of Il-10 in hepatocytes is especially hard to understand because this interleukin is almost specifically secreted by immune cells and not for parenchymal liver units. So, this fact would be responsible for the dispersion of the results (being almost within the limit of quantification of the method (normally in the picograms range), the interpretation of data relative to IL-10 should be reinterpreted and redrafted carefully.
Answer: Thank you very much for the professional comments, and it really help us to improve the quality of our manuscript.
- In our study, we measured the levels of IL-10 in animal serum samples, not liver tissue.
- Previous studies also found the increased levels of TNF-α and IL-10 in serum or liver tissue in obesity-related NAFLD rat and mouse models established by high-fat diet or monosodium glutamate (MSG) injection 【Life Sci . 2021 Feb 1;266:118868.】【Biomed Pharmacother . 2017 Jun;90:608-614.】. We copy some of the data in published paper as following: (please see attached file.)
- We supplemented the discussion about important cytokine marker of inflammation, TNF-α and IL-10, in revised manuscript as following. Please refer to line 740 to 752, which was marked with red fonts.
TNF-α and IL-10 play a bridging role in insulin resistance (IR) in NAFLD. It has been found that long term of high-fat diet can trigger the transformation of liver Kupffer cells (KC) into M1 phenotype, resulting in the up-regulated expression of pro-inflammatory cytokines such as TNF-α, which could further aggravate fat accumulation in hepatocyte, thus affecting the fatty acid oxidation, TG accumulation and IR of hepatocytes【J Biol Chem . 2010 Sep 24;285(39):29965-73.】【Rev Endocr Metab Disord . 2016 Mar;17(1):29-39. 】.Fujisaka et al. 【Diabetes. 2009 Nov;58(11):2574-82.】found that the number of M1-type macrophages and M1/M2 are closely related to the pathogenesis and development of IR. High fat diet can induce adipose tissue macrophages (ATMs) to polarize towards M1-type macrophages. The anti-inflammatory Th2 cytokine IL-10 can increase the number of M2-type macrophages, lead to congenital immune imbalance, and eventually lead to hepatocyte steatosis.
Some subsections regarding methodology show a lack of proper citations while not enough details for reproduction are provided. This is the case of subsection 2.2.2. Determination of the liver index and spleen index. Beyond this, in this section, the formula included is mis expressed and should be correctly applied to both the liver and spleen determinations.
Answer: Thank you very much for the professional comments.
In the case of subsection 2.2.2, the methodology was cited a publish paper 【PLoS One, 2021. 16(8): p. e0256594.】, and the calculation formula was also corrected as following: organ index = [weight organ (g) /weight body (g)] × 100 %. Please refer to line 134 and 135 in revised manuscript, which was marked with red fonts.
Additional information (description and discussion) on the organ index is required.
Answer: Thank you very much for the professional comments.
The description and discussion on the organ index were supplemented in revised manuscript as followings, which were marked with red fonts. Please refer to discussion section line 700 to 710.
“Clinical data showed that NAFLD patients disclose a relatively high prevalence of hepatomegaly and splenomegaly, a common feature of patients with advanced chronic liver disease, such as viral hepatitis and cirrhosis, which is probably due to haemody-namic alteration of portal flow toward hypertension, as a consequence of liver fibrosis [22].
In animal experiment, organ index can be used to evaluate the overall organ status. Abnormal increasing of parenchymatous organ index is usually associated with the some damage to organ, which exhibits swell or accrete [20]. In this study, we found that quinoa treatment could effectively inhibit the increasing of indexes of liver and spleen. The may partially attribute to its beneficial role in the prevention of NAFLD, such as ameliorating hepatic adiposopathy, rescuing inflammatory reactions and oxidative stress, and regulation of the gut microbiota dysbiosis.”
Concerning the modulation of microbiota, the description should be carefully reviewed. Especially for stressing adequately those effects of relevance. In this regard, almost all treatments and target species monitored produced no such desirable effects being closer to the High-fat group than the Control group. In this sense, the statements should be modulated indicating the limited capacity of quinoa to act on this risk factor for NAFLD, also, this information should be included in the conclusions section.
Answer: Thank you very much for the professional comments.
We have made correction in discussion and conclusion section in revised manuscript. Please refer to line 767 to 770, line 793 to 796, line 802 to 804, and line 813 to 815, which were marked with red fonts.

Round 2
Reviewer 2 Report
The authors have addressed almost all concerns about the previous version of the MS. So, in my opinion, the article can be accepted for publication in its present form